TOXIFY: a deep learning approach to classify animal venom proteins

Cole T. Jeffrey coleti16@students.ecu.edu
Brewer Michael S. brewermi14@ecu.edu
Department of Biology, East Carolina University , Greenville , NC , United States of America
Paci Emanuele
Electronic publication date: 2019 Jun 28
Publication date: 2019
Volume: 7
Electronic Location ID: e7200
Received 2019 Jan 18; Accepted 2019 May 29
Copyright: ©2019 Cole and Brewer
Copyright year: 2019
Copyright holder: Cole and Brewer
License: This is an open access article distributed under the terms of the Creative Commons Attribution License, which permits unrestricted use, distribution, reproduction and adaptation in any medium and for any purpose provided that it is properly attributed. For attribution, the original author(s), title, publication source (PeerJ) and either DOI or URL of the article must be cited.
License URL: https://creativecommons.org/licenses/by/4.0/

Keywords: Venom, Deep learning, Protein classification, Transcriptome, Proteome

Funding: National Science Foundation Graduate Research Fellowship East Carolina University Department of Biology This work was supported by a National Science Foundation Graduate Research Fellowship to T. Jeffrey Cole and the East Carolina University Department of Biology. The funders had no role in study design, data collection and analysis, decision to publish, or preparation of the manuscript.

==============================
In the era of Next-Generation Sequencing and shotgun proteomics, the sequences of animal toxigenic proteins are being generated at rates exceeding the pace of traditional means for empirical toxicity verification. To facilitate the automation of toxin identification from protein sequences, we trained Recurrent Neural Networks with Gated Recurrent Units on publicly available datasets. The resulting models are available via the novel software package TOXIFY, allowing users to infer the probability of a given protein sequence being a venom protein. TOXIFY is more than 20X faster and uses over an order of magnitude less memory than previously published methods. Additionally, TOXIFY is more accurate, precise, and sensitive at classifying venom proteins.

Introduction

Venom is a complex trait utilized across numerous animal phyla and is an ideal source for novel drug discoveries with millions of peptides highly specific for therapeutically relevant ion channels, G-protein-coupled receptors, enzymes, and transporters (Prashanth, Hasaballah & Vetter, 2017). Despite recent advances in venom gland proteomics and transcriptomics that have yielded millions of potential venom peptides (Fry et al., 2009), the biological function of only approximately 8,000 entities have been empirically verified and deposited into UniProt to date. In lieu of empirical evidence, venom proteins are often identified via BLAST searches, which can be a problematic approach for two primary reasons. First, venom proteins frequently evolve via gene duplications and subsequent neofunctionalization and subfunctionalization of non-toxic physiological proteins, rendering BLAST unable to possess the sensitivity needed to distinguish venomous paralogs (Hargreaves et al., 2014; Duda & Palumbi, 1999). Second, venom peptides are often highly diverged to the extent that BLAST searches yield no strong matches (Linial, Rappoport & Ofer, 2017). Gacesa, Barlow & Long (2016) attempted to alleviate these difficulties through their software ToxClassifier with Generalized Linear Models (GLM) and by training a support vector machine (SVM) on venom and non-venom proteins encoded with Hidden Markov Models (HMM) of tox-bit motifs via BLAST searches against a positive venom database (Gacesa, Barlow & Long, 2016). However, the SVM-based approach utilized by ToxClassifier is restricted to linear representations, which may limit performance in regard to classifying protein sequences (Byvatov et al., 2003). Here we present toxify, a deep-learning approach to distinguish animal venom proteins from non-toxic proteins by training neural networks on protein sequences encoded as numerical vectors.

Methods

Datasets

To allow for a proper comparison between ToxClassifier and toxify, the training datasets for toxify comprised only protein sequences from UniProtKB that were uploaded/available pior to June 2016, when ToxClassifier was published. Protein sequences uploaded to UniProtKB between June 2016 and October 2018 were not included as training data for either ToxClassifier or toxify and were used as benchmark comparisons between the two methods. Datasets were obtained using the following two procedures.

1. To train our models to classify venom proteins, we constrained our training sets to only include verified venom proteins from Swiss-Prot. This dataset, referred henceforth as “positive”, was constructed using the following search terms (annotation:(type:“tissue specificity” venom)). This resulted in a total of 6,133 venom protein sequences.

2. “Negative” datasets comprised 50,000 random, non-venom proteins from Swiss-Prot using the following search term (NOT annotation:(type:“tissue specificity” venom) AND reviewed:yes).

Due to venom proteins generally being low mass and relatively short (e.g., <30 AA), only proteins containing ≤500 amino acids were included in the final training dataset. This brought the size of the positive dataset down to a total of 4,808 proteins and the negative dataset to 32,391 proteins. Training data consisted of a random 80% subset of the positive and negative sequences, and the remaining 20% was set aside for model validation. Less than 5% of the dataset contained sequence redundancy, which is an artifact of the databasing procedure in SwissProt.

Sequence encoding

In order to encode proteins as numerical representations, the amino acid sequence for each protein was inscribed as a 5x500 matrix with values corresponding to each of the five Atchley factors per amino acid in the protein (Atchley et al., 2005). Atchley factors were chosen because they statistically summarize 500 AA attributes that reflect polarity, secondary structure, molecular volume, codon diversity, and electrostatic charge. Matrices of proteins containing fewer than 500 amino acids were padded with zeros to fill the remainder of the matrix, which do not affect the resulting models and predictions.

Sequence classifier

Recurrent Neural Networks (RNN) are an ideal tool for classifying ordered sets of items, such as amino acid sequences, because they specify hidden states that depend on the input as well as the prior hidden state. Gated Recurrent Units (GRUs) are a high-performing RNN that have gained popularity since being introduced by Cho et al. (2014), due to faster performance over traditional Long Short-Term Memory approaches. Using TensorFlow v1.8.0 libraries as the back-end (Abadi et al., 2016), we constructed a venom protein classifier using GRU with 270 hidden units with a learning rate of 0.01. Training occurred for 50 epochs, and the training accuracy and training loss in accuracy (from a logit cost function) were recorded at every 2nd epoch. The trained model was then used to calculate the probability that a given protein should be classified as an animal venom (Fig. 1).

Figure 1 Workflow diagram for toxify, including preprocessing of training data, filtering by size and zero padding, converting numeric vectors as Atchley factors, and training on neural network using gated recurrent units.

Benchmark against ToxClassifier

While other machine learning approaches exist, such as ToxinPred (Gupta et al., 2013), ClanTox (Naamati, Askenazi & Linial, 2009) and ToxClassifer are the only ones that support large proteomic datasets, so we chose them to benchmark against toxify. Test data used to compare performance of toxify to ToxClassifier included all protein sequences from Swiss-Prot with ≤500 amino acids that have been uploaded since the publication of ToxClassifier (2016–2018) and not used in the training set of either toxify, Toxclassifier, or ClanTox. These were split into a positive dataset with all 274 verified venom protein sequences and a negative dataset with 274 randomly selected and verified non-venom protein sequences.

ClanTox scores toxins on a scale of −1 to 1, with -1 being unlikely to be a venom and 1 being very likely to be a venom. The overall venom probability for ClanTox was calculated by normalizing the values to range between 0 and 1. ToxClassifier uses nine SVM and GLM models and provides as output a nine dimensional vector of zeros and ones indicating the prediction of whether or not the protein is a venom from each model. The probability for ToxClassifier was calculated as the average for all models. Contrarily, the probability a protein is a venom from toxify was determined as the probability calculated for each protein by the trained network. Thus, the sum of these probabilities gives an approximation of the total number of predicted toxins in a dataset, and serves as the basis for bench-marking the performance of the different models with the same following criteria as Gacesa, Barlow & Long (2016).

1. True positive (TP): This is the approximate number of positive sequences that were correctly predicted to be venom proteins. It is calculated as the product of the mean venom probability of the positive dataset and the total number of positive sequences.

2. True negative (TN): This is the approximate number of negative sequences that were correctly predicted to be non-venom proteins. This is calculated as the product of the mean venom complementary probability of the negative dataset and the total number of negative sequences.

3. False positive (FP): This is the approximate number of negative sequences that were incorrectly predicted to be venom proteins. This is calculated as the product of the mean venom probability of the negative dataset and the total number of negative sequences.

4. False negative (FN): This is the approximate number of positive sequences that were incorrectly predicted to be non-venom proteins. This is calculated as the product of the mean venom complementary probability of the positive dataset and the total number of positive sequences.

5. Accuracy (ACC): This is the proportion of proteins that were correctly predicted to be either a venom protein or a non-venom protein. This was calculated using the following expression; ACC=TP+TNTP+TN+FP+FN

6. Specificity (SPEC): This is the proportion of proteins that were correctly classified as a non-venom protein. This was calculated using the following expression;

SPEC=TNTN+FP

7. Sensitivity (SENS): This is the proportion of proteins that were correctly classified as a venom protein. This was calculated using the following expression; SENS=TPTP+FN

8. Balanced accuracy (BACC): This is the average of specificity and sensitivity, which was calculated using the following expression; BACC=SPEC+SENS2

9. Negative predictive value (NPV): This is the proportion of negative sequences that were classified as true negatives, which was calculated using the following expression;

NPV=TNTN+FN

10. Positive predictive value (PPV): This is the proportion of positive sequences that were classified as true positives, which was calculated using the following expression;

PPV=TPTP+FP

11. F-score (F1): This is the harmonic mean of precision and sensitivity, which was calculated using the following expression; F1=2×TP2×TP+FP+FN

12. Matthew’s Correlation coefficient (MCC): This is a measure of the correlation between observed and predicted values (Matthews, 1975; Powers, 2011), which was calculated using the following expression;

MCC=TP×TN+FP×FNTP+FP×TP+FN×TN+FP×TN+FN

These analysis were carried out on the same Linux computer (Ubuntu 18.04; Dell PowerEdge-R910; 48 threads; 1 Tb RAM). The elapsed real time between invocation and termination of each process by the CPU was recorded in seconds(s), and the maximum resident set size of memory allocation (MEM) was recorded in megabytes (MB).

Results

Validation

The overall accuracy of the model was assessed every 2nd epoch during training as recorded by the loss of accuracy, the training accuracy, and the test dataset accuracy (Fig. 2). At the end of training, the model approached 99.9% training accuracy and 97.4% accuracy on the test dataset.

Figure 2 Accuracy progression for RNN as training progressed for both training and test datasets.

Benchmark against ClanTox and ToxClassifier

Of the 274 venom and non-venom proteins uploaded to Swiss-Prot since the publication of ToxClassifier, toxify correctly classified 37% more venom proteins than ToxClassifier. toxify also correctly classified 96.0% of non-venom proteins, which is only 2.7% less than the amount correctly classified by ToxClassifier. ToxClassifier and toxify outperformed ClanTox on all metrics. Further, toxify completed these predictions an order of magnitude faster than ToxClassifier, and used an order of magnitude less memory (Table 1).

Table 1 Benchmark metrics for toxify compared to clantox and Toxclassifier.

The top portion shows averages and percentages (in parenthesis) of true and false positives (TP & FP) as well as true and false negatives (TN & NF). Additionally, the proportions for accuracy, specificity, sensitivity, balanced accuracy, negative predictive value, positive predictive value, F-score, and Matthew’s correlation coefficient are listed. The bottom portion shows computational performance in terms of CPU time in seconds and memory usage in megabytes. Asterisks indicate metrics in which toxify outperformed.

	clantox	ToxClassifier	toxify	
TP*	147.8 (54.2%)	152.8 (55.8%)	209.6 (76.5%)	
TN	223.4 (81.8%)	270.2 (98.6%)	263.0 (96.0%)	
FP	50.6 (18.5%)	3.8 (1.4%)	11.0 (4.0%)	
FN*	126.2 (46.2%)	121.2 (44.2%)	64.4 (23.5%)	
ACC*	0.68	0.77	0.86	
SPEC	0.82	0.99	0.96	
SENS*	0.54	0.56	0.76	
BACC*	0.68	0.77	0.86	
NPV*	0.64	0.69	0.80	
PPV	0.74	0.98	0.95	
F1*	0.63	0.71	0.85	
MCC*	0.37	0.60	0.74	
CPU (s)*	NA	100.18	4.05	
MEM (MB)*	NA	6,824	293	

Implementation

Sensitivity against non-toxic homologs

To ensure that toxify is able to discern between toxins and homologous physiological proteins, toxify classified the top hits from a BLASTP search of the remaining 502,000 proteins in Swiss-Prot (not included in the training dataset) against the positive dataset keeping hits with an e-value cutoff of 1 × 10−6. toxify correctly assigned at least a 90% non-venom probability to 1,961 out of 2,183 (89.8%) of the non-toxic homologs, which is only 6.2 percentage points lower than toxify’s true negative rate when calculated for non-toxic non-homologous proteins, and 5.2 percentage points lower than the accuracy reported by ToxClassifier on a similar dataset denoted by Gacesa, Barlow & Long (2016) as their “hard” dataset.

Case usage

Venom proteins from highly studied venomous animals made up the bulk of the training data. In order to assess the robustness of toxify to other venomous animals, we classified a recently discovered venom protein expressed by a robber fly (Diptera: Asilidae: Machimus arthriticus), which belongs to a lineage of venomous flies whose venoms had not been previously studied and were not found in Swiss-Prot before our cutoff date (Drukewitz et al., 2018). The neurotoxin described by Drukewitz et al. (2018) as asilidin1 was discovered through venom gland transcriptomics and venom proteomics, and its activity as a neurotoxin was verified via injection of the synthesized peptide into a honey bee (Apis mellifera), which caused subsequent paralysis. toxify classified asilidin1 as a venom protein with a probability of 0.99.

While asilidin1 is the only toxin that Drukewitz et al. (2018) empirically verified to have toxigenic effects, they also discovered 169 additional putative toxins. We provided those sequences as well as 170 random complete coding seqeunces from the body tissue transcriptome of Machimus arthriticus as input. toxify classified a total of 50 out of the 169 putative toxins described by Drukewitz et al. (2018) as being toxins, but only 10 of the 170 randomly sampled non-venom proteins from the same species. This suggests that the predictive power of the models used by toxify is reduced in venomous lineages not included in the training datasets, nevertheless the true positive rate is five times higher than the false positive rate.

Discussion

In this paper, we proposed toxify, the first animal toxin classifier that utilizes deep learning. While other classifiers use slower BLAST-based methods to encode proteins as a set of features as training input for SVM models (Naamati, Askenazi & Linial, 2009; Gacesa, Barlow & Long, 2016), toxify quickly encodes amino acids as a 5-dimensional vector of Atchley factors and makes use of the sequential nature of proteins via neural networks with GRUs to classify sequences. While toxify had a slightly poorer false positive rate and specificity scores when compared to ToxClassifier, it achieved a better accuracy, sensitivity score, F-score, and Matthew’s correlation coefficient—while achieving speeds and memory efficiency an order of magnitude better than ToxClassifier.

There are two obvious applications for toxify. First of all, the prediction module of toxify may be used to quickly classify large venom protein datasets. These datasets may be in the form of venom gland transcriptomes or venom proteomes and can stem from a wide variety of venomous taxa. Users may also find toxify to be a helpful screening tool for annotation pipelines such as Venomix (Macrander et al., 2018). Secondly, the training module of toxify has been setup for users to easily retrain models on custom protein datasets, while easily adjusting hyper-parameters such as the number of training epochs, learning rate, and unit size for the GRUs.

While we were able to demonstrate toxify’s utility in classifying venom proteins across the phylogenetic diversity of eumetazoans, improvements in the overall accuracy between taxa can still be made. Balancing the training dataset is still an issue due to verified venom protein sequences being outnumbered by non-venom proteins by several orders of magnitude. Improvements in the accuracy of the current iteration of toxify will be accomplished by empirically verifying the functionality of additional venoms from an ever increasing number of sequenced venom proteins from a broader array of venomous taxa. In the meantime, a tool like toxify in tandem with ToxClassifier (Gacesa, Barlow & Long, 2016), ToxinPred (Gupta et al., 2013), ClanTox (Naamati, Askenazi & Linial, 2009), and Venomix (Macrander et al., 2018) may expedite the process of narrowing down potentially novel venoms to be empirically verified and added to repositories such as Swiss-Prot.

We thank Chris Cohen for editing and improving the manuscript and two reviewers for their helpful comments.

Additional Information and Declarations

Competing Interests

Author Contributions

Data Availability

The authors declare there are no competing interests.

T. Jeffrey Cole conceived and designed the experiments, performed the experiments, analyzed the data, prepared figures and/or tables, authored or reviewed drafts of the paper, approved the final draft.

Michael S. Brewer conceived and designed the experiments, analyzed the data, contributed reagents/materials/analysis tools, prepared figures and/or tables, authored or reviewed drafts of the paper, approved the final draft.

The following information was supplied regarding data availability:

Source code is hosted on GitHub and can be found at https://www.github.com/tijeco/toxify.

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
