# Peer review of "TOXIFY: a deep learning approach to classify animal venom proteins"

_PeerJ, doi:10.7717/peerj.7200_

## Round 0.1 · original submission · Major Revisions

The reviews are generally positive and provide some suggestions for improving the manuscript. A decision of "major revision" gives you the opportunity to address all the reviewers' comments. I would also like to add the suggestion to provide a rationale for using deep learning instead of other statistical tools, and in general provide more details on the methods used.

·

Basic reporting

The authors present a novel software package TOXIFY, for inferring the probability of being a venom protein which is 20X faster and uses less memory than previously published methods. Toxify is a nice technical tool. Professionally written. The motivation and results are clearly presented.
Importantly, the authors share the tool and code used in an easy to use CMD interface/tool that is commendable.

Experimental design

1. The design is valid and well presented. However, The authors used TOXIFY in tandem with TOXCLASSIFIER. Testing with respect to other ML method is needed to better understand the performance in a more general sense. It is needed for supporting the conclusion stated in the abstract (Lines 15-16). At present, this statement should be written more precise.

2. The degree of sequence redundancy in the all the training, test and the showcase were not mentioned. It is relevant to the final performance

Validity of the findings

The findings seem appropriate. However, the datasets available on the GitHub page do not match the numbers in the article. Specifically, in the Github, the training set holds 12,462 rows/sequences, with 367 in the benchmark test data. The article states other numbers (4,808 and 32,391 for positive and negative proteins. Authors should clarify the discrepancy in numbers.

Additional comments

Few minor comments for improving the presentation
a. Line 28, Numerous studies support this claim and citations are missing (e.g. doi: 10.3390/toxins9110350).
b. Lines 33-34: need a small revision, as SVMs can capture nonlinear relationships using kernels, e.g. RBF kernels, and/or feature transformations of the input space.
c. Line 171: The concept of transforming sequences into a rich set of features rather than the slow BLAST alignment was presented. Citation is missing (e.g. doi.org/10.1093/bioinformatics/btv345 and others.
d. Table 1. Please explain the asterisks
e. Lines 166-168 need a reference.

·

Basic reporting

The basic reporting in this article is good.

Experimental design

I have issues with the experimental design in sections 3.3.2, and 3.3.3 as detailed in General Comments below.

Validity of the findings

Addressing the experimental design issues I have raised will increase the confidence of the reader in the validity of the findings.

Additional comments

Review of PeerJ-34316-v0

Disclaimer
Please note that I am a molecular biologist and toxinology researcher with very limited knowledge of deep learning and associated fields, and this review is confined to those aspects I am capable of addressing.

General comments
This article describes the development and testing of TOXIFY, a deep learning algorithm aiming to classify toxins or non-toxins based on their primary structures. The problem addressed is a worthwhile one, and the manuscript has been carefully prepared. From the data presented, the authors have produced an algorithm displaying considerable increases in both accuracy and speed compared to both traditional annotation methods and previous machine learning approaches. My main issues with the manuscript relate to how the algorithm has been tested, and how those tests are presented in the manuscript. These are listed below. I recommend the manuscript is suitable for publication after revisions to address these issues and/or the authors have responded to these issues raised.
• Section 3.3.2: I am not sure this section demonstrates anything meaningful. Am I right in thinking that all it shows is that Toxify identifies more ORFs as toxins than BLAST searches do? If so, this may or may not be a desirable feature, depending on whether these many thousands of additional sequences actually represent toxins or not. From what is presented, there is no reason to think that this is the case except the assertion on lines 157–159 that “Because many additional venoms are known to be present in these taxa beyond the number indicated by BLAST searches, this highlights TOXIFY ’s greater predictive power compared to BLAST-based approaches.” A more convincing demonstration would test the ORFs in this way and then compare the results to some kind of empirical classification of if the protein sequences were venom or non-venom, most ideally if they can be detected in venom by mass spectrometry. Surely this would be possible for one of these or another well-characterised species?
• Section 3.3.3: I also have several problems with this section: (a) that the test consists of a single ‘positive’ sequence and no negatives; and (b) that this sequence appears to belong to a protein superfamily (inhibitor cystine knots) that is represented not just in asilid venoms but also the venoms of numerous other animal groups, particularly spiders and cone snails. This means that although this test does indeed support the applicability of Toxify to ‘other venomous animals’ not included in the training group, the scope of this may be considerably more limited than what is inferred. How would Toxify perform on the numerous other protein families recently discovered in asilid venom? Ideally, I would like to see the study measure (or at least discuss the issues surrounding) how ‘novel venomous taxa’ vs. ‘novel venom protein families’ affect Toxify’s performance.
• Table 1: This table compares the performance of ToxClassifier and Toxify on the negative and positive datasets described on lines 79–81: “a positive dataset with all 274 verified venom protein sequences and a negative dataset with 274 randomly selected and verified non-venom protein sequences.” Tested on this dataset, Toxify is in general more happy to classify things as venom proteins, and has a substantial increase in true positive (TP) and decrease in false negative (FN) with only very slight increase in false positive (FP) and decrease in true negative (TN). However, the ToxClassifier algorithm seems to have been designed and tested on quite different datasets. The negative dataset used to generate Table 1 seems to more closely resemble the ‘easy’ dataset used in the Gacesa et al. 2016 article (10.7717/peerj-cs.90), whereas the ‘moderate’ dataset in this 2016 study consisted of sequences with blast homology to verified toxins but that were verified non-toxins, more similar to that described in 3.3.1. How would Toxify compare to ToxClassifier on this more difficult type of dataset?

Other minor comments
• Lines 79–81: Can you tell us more detail of how the sequences in the negative test dataset were verified as non-venom?
• Lines 88–99: Why can we not tell the exact number of sequences of each type (TP, TN, FP, FN)?

Reviewer 3 ·

Basic reporting

The basic report is good.

Experimental design

The purpose of the experiment is clearly explained. The data used is presented adequately.

Validity of the findings

I have some comments about the representation of the algorithm I included in the general comment.

Additional comments

I am statistician and including some equations explains how TOXIFY algorithm tackles the classification of toxic sequences will enhance the work and make the comparison with existing algorithm more visible.

---

## Round 0.2 · accepted · Accept

The revised manuscript is acceptable for publication.

# ·

Basic reporting

The basic reporting in this manuscript is now good.

Experimental design

The experimental design seems appropriate to the scope of the study.

Validity of the findings

I think the validity and context of the findings is substantially improved in this revised version.

Additional comments

Thankyou for carefully considering the revisions and comments and addressing them. I am satisfied that the points I raised in response to the first submission of this manuscript have been sufficiently addressed and recommend that the article is now suitable for publication.